# The T Cell Receptor (TRB) Locus in *Tursiops truncatus*: From Sequence to Structure of the Alpha/Beta Heterodimer in the Human/Dolphin Comparison

**DOI:** 10.3390/genes12040571

**Published:** 2021-04-14

**Authors:** Giovanna Linguiti, Sofia Kossida, Ciro Leonardo Pierri, Joumana Jabado-Michaloud, Geraldine Folch, Serafina Massari, Marie-Paule Lefranc, Salvatrice Ciccarese, Rachele Antonacci

**Affiliations:** 1Department of Biology, University of Bari, “Aldo Moro”, 70124 Bari, Italy; giovanna.linguiti@uniba.it (G.L.); rachele.antonacci@uniba.it (R.A.); 2IMGT^®^, the international ImMunoGeneTics information system^®^, Laboratoire d’ImmunoGénétique Moléculaire (LIGM), Institut de Génétique Humaine (IGH), 34000 Montpellier, France; sofia.kossida@igh.cnrs.fr (S.K.); joumana.michaloud@igh.cnrs.fr (J.J.-M.); geraldine.folch@igh.cnrs.fr (G.F.); 3Department of Biosciences, Biotechnologies and Biopharmaceutics, University of Bari, “Aldo Moro”, 70124 Bari, Italy; ciro.pierri@uniba.it; 4BROWSer S.r.l., 70124 Bari, Italy; 5Department of Biological and Environmental Science and Technologies, University of Salento, 73100 Lecce, Italy; sara.massari@unisalento.it

**Keywords:** T cell receptor, dolphin genome, TRB locus, TRBV, TRBJ, TRBD and TRBC genes, TRA and TRB gene expression analysis, multiple sequence alignments (MSA), 3D modelization, IMGT

## Abstract

The bottlenose dolphin (*Tursiops truncatus*) belongs to the Cetartiodactyla and, similarly to other cetaceans, represents the most successful mammalian colonization of the aquatic environment. Here we report a genomic, evolutionary, and expression study of *T. truncatus* T cell receptor beta (TRB) genes. Although the organization of the dolphin TRB locus is similar to that of the other artiodactyl species, with three in tandem D-J-C clusters located at its 3′ end, its uniqueness is given by the reduction of the total length due essentially to the absence of duplications and to the deletions that have drastically reduced the number of the germline TRBV genes. We have analyzed the relevant mature transcripts from two subjects. The simultaneous availability of rearranged T cell receptor α (TRA) and TRB cDNA from the peripheral blood of one of the two specimens, and the human/dolphin amino acids multi-sequence alignments, allowed us to calculate the most likely interactions at the protein interface between the alpha/beta heterodimer in complex with major histocompatibility class I (MH1) protein. Interacting amino acids located in the complementarity-determining region according to IMGT numbering (CDR-IMGT) of the dolphin variable V-alpha and beta domains were identified. According to comparative modelization, the atom pair contact sites analysis between the human MH1 grove (G) domains and the T cell receptor (TR) V domains confirms conservation of the structure of the dolphin TR/pMH.

## 1. Introduction

The bottlenose dolphin (*Tursiops truncatus*) and the other cetaceans phylogenetically belong to the Cetartiodactyla clade, which includes Cetacea and Artiodactyla [1,2]. The divergence of cetaceans from their terrestrial ancestors about 54 million years ago [3] resulted in a complete adaptation to aquatic life. In addition to anatomical and physiological innovations required for life in water, cetaceans must have been confronted with challenges from changing environmental pathogens while they transitioned from land to sea [4,5]. These challenges exerted intensified selection pressure on the whole genomes of cetacean lineage, including the Delphinidae family [6], as well as on gene families related to the immune system [7].

αβ or γδ T cell receptor (TR) heterodimers are members of the immunoglobulin superfamily (IgSF). The organization of each gene family encoding for the α, β, γ and δ chains comprises distinct genomic loci that are named TRA, TRB, TRG, or TRD based on the different TR chains [8]. Through a random process of DNA rearrangement that implies somatic recombination of V (variable), D (diversity), and J (joining) genes for the β and δ chains, and V and J genes for the α and γ chains, each gene family encodes the variable domain of its own chain, to obtain an effective repertoire of TR [8]. The resulting rearranged V (D) J-region, after transcription, is spliced to the C (constant) gene that encodes the constant region of each chain of the receptor [8]. The variable domain, at the N-terminal end of each chain, is composed of seven distinguishable regions: four framework regions (FR-IMGT) and three hypervariable loops or complementarity-determining regions (CDR-IMGT). The first two CDR loops, CDR1-IMGT and CDR2-IMGT, are encoded by the V gene, while the third CDR loop (CDR3-IMGT) reflects the ability of the V gene to rearrange to any (D) J gene [8]. Each TR locus contains the V, D, J, and C genes as multigene subfamilies, and the total number of their gene members changes in different species. Information about the potential for the adaptative immune response against antigens within a given species can be estimated by the identification of its TR germline repertoire.

Recent studies enabled us to determine and characterize the genomic organization of the dolphin TRG, TRA, and TRD loci [9]. Comparative studies showed that the dolphin TRG locus is the simplest and the smallest among the mammalian TRG loci identified to date [10,11,12] and is organized in a single V-J-C cassette comprising only 2 TRGV, 3 TRGJ genes, and a single TRGC gene [9].

In particular, the comparative analysis highlighted the overall organization of the dolphin TRG locus which resembles the structure of the TRGC5 cassette, the ancestral unit that gave rise to reiterated duplications in ruminants TRG locus [13,14]. Dolphin genome assembly, although fragmented and incomplete, confirmed that the TRD genes are clustered within the TRA locus, as in *Homo sapiens* [8] and other eutherians and in birds, and revealed a conserved structure and organization with respect to the artiodactyls and the others terrestrial mammals analyzed so far [8,15,16,17]. Regarding expression data, the dolphin TRG and TRA/TRD potential repertoire provided evidence for an unusual ratio of productive/unproductive transcripts which arise from the TRG V-J gene rearrangement and allowed the prediction of the most likely and most stable γδ pairing for a “public” gamma delta TR repertoire [9].

The aim of this work was to provide a scenario on the T cell receptor loci in *T. truncatus* analyzing the germline and expressed TRB repertoires. Thus, with the release of the most recent dolphin genome assembly (mTurTru1.mat.Y), we deduced the genomic structure of the dolphin TRB locus, we characterized and annotated its genes, and we have highlighted its uniqueness in comparative analysis with other artiodactyls. Expression analysis allowed us to evaluate the contribution of each V, D, J, single gene to the dolphin TRB repertoire. Starting from the TRA and TRB cDNA sequences, we were able to build a 3D human/dolphin comparative modeling of the *T. truncatus* TR alpha/beta in complex with major histocompatibility class I (MH1) protein and beta-2 microglobulin (B2M) and to identify the resulting interaction energy between amino acids sidechains in the CDR-IMGT (IMGT^®^ [18] http://www.imgt.org, accessed on 10 February 2021) of the dolphin TR/MH structure.

## 2. Materials and Methods

### 2.1. Genome Analyses

To analyze the TRB locus of the bottlenose dolphin (*T. truncatus*), the mTurTru1.mat.Y assembly of the whole genome shotgun sequence (BioProject ID: PRJNA625792) [19] obtained by Illumina NovaSeq sequencing technology, was searched using the BLAST algorithm [20].

A sequence of 277,931 bp (gaps included) was retrieved directly from a unique genomic scaffold mapped on chromosome 9 (NW_022983121.1) of the reference sequence NC_047042.1, identical to CM022282.1, available at NCBI from 93263345 to 93541276. A flowchart of the overall experimental setting and analyzes is reported in Appendix A.

The *T. truncatus* TRB locus is in forward (FWD) orientation on chromosome 9. In particular, the analyzed region comprises the MOXD2 (monooxygenase DBH like 2, pseudogene) and EPHB6 (EPH receptor B6) genes, already annotated within the scaffold and flanking respectively, at the 5′ and 3′ ends of the TRB locus.

The genomic sequences of the TRB locus of humans (*Homo sapiens*) [8,21,22,23] and of Artiodactyla i.e., Arabian camel or dromedary (*Camelus dromedarius*), pig (*Sus scrofa*), and sheep (*Ovis aries*) [24,25,26,27] were used against the dolphin genome sequence to identify, based on homology by the BLAST program, the corresponding genomic TRBV, TRBD, TRBJ, and TRBC genes.

The beginning and the end of each coding exon were identified with accuracy by the presence of splice sites or of recombination signal (RS) sequences (V-RS, 5’D-RS, and 3’D-RS, J-RS for TRBV, TRBD, and TRBJ genes, respectively). Genes’ TRB location is provided in Appendix A. Sequence comparison has allowed the identification and characterization of the dolphin protease genes (TRY). TRY genes location and the MOXD2 and EPBH6 flanking TRB locus genes, are provided in Appendix A.

Computational analysis of the dolphin TRB locus was conducted using two different programs: RepeatMasker for the identification of low complexity regions and genome-wide repeats [28] (http://repeatmasker.org, accessed on 10 February 2021) and Pipmaker [29] (http://pipmaker.bx.psu.edu/pipmaker/, accessed on 10 February 2021) for the alignment of the dolphin sequence with human (GenBank accession number L36092), goat (GenBank accession number NC_030811.1), pig (GenBank accession number NC_0101460.4), and Arabian camel (GenBank accession number NW_011591622.1) counterparts. RepeatMasker screens DNA sequences for interspersed repeats and low complexity DNA sequences.

### 2.2. Classification of the Dolphin TRB Genes

Considering the percentage of nucleotide identity of the genes with respect to human [8,21,30] and the other mammalian species and based on the genomic position within the locus, each TRB gene was classified, and the nomenclature was established according to the IMGTconcepts of classification, approved by HGNC, VGNC, NCBI and the IUIS Nomenclature Committee for immunoglobulin (IG) and TR of all vertebrate species with jaws (gnathostomata) from fish to humans [8,23,31] (Appendix A). The functionality of the *V*, *D*, *J*, and *C* genes, based on the IMGT Scientific chart rules [18], was predicted through the manual alignment of sequences adopting the following parameters: (a) identification of the leader sequence at the 5′ of the TRBV genes; (b) determination of proper RS located at 3′ of the TRBV (V-RS), 5′ and 3′ ends of the TRBD (5′D-RS and 3′D-RS) and 5′ of the TRBJ (J-RS), respectively; (c) determination of conserved acceptor and donor splicing sites (IMGT^®^ [18], IMGT Education > IMGT Aide-mémoire > Splicing sites); (d) estimation of the expected length of the coding regions; (e) absence of frameshifts and stop codons in the coding regions of the genes (IMGT^®^ [18], IMGT Scientific chart > 1. Sequence and 3D structure identification and description > IMGT functionality).

The criterion that sequences with a nucleotide identity of more than 75% in the V-region belong to the same subgroup was adopted to assign the TRBV genes to 20 different subgroups. The percentage of nucleotide identity has been identified by using the Clustal Omega alignment tool, which is available at the EMBL-EBI website (http://www.ebi.ac.uk/, accessed on 10 February 2021).

The TRBD, TRBJ, and TRBC genes were annotated, according to the similarity with the other artiodactyl species and their specific organization in three D-J-C clusters, numbered C1, C3, and C2 from 5’ to 3’ in the locus [25,26,27]. Each TRBJ gene of the TRBJ1, TRBJ2, and TRBJ3 sets was designed by a hyphen and a number corresponding to their position in the cluster. They were all predicted to be functional, except for the TRBJ1-5 and TRBJ3-6 which are pseudogenes owing to STOP-CODON and TRBJ1-2 (noncanonical J- NONAMER), TRBJ1-4 (noncanonical J-HEPTAMER), and TRBJ3-2 (nonconserved J-MOTIF) classified as an open reading frame (ORF) (Appendix A).

### 2.3. Phylogenetic Analyses

The TRBV genes used for the phylogenetic analysis were retrieved from the following sequences deposited in the GEDI (for GenBank/ENA/DDBJ/IMGT/LIGM-DB) databases: NW_011591622, NW_011593440, NW_011591151 (*Camelus dromedarius* TRB locus contig as previously characterized [26,27]), NC_010460 (*Sus scrofa* TRB locus contig as previously characterized [24]); and NC_030811.1 (*Capra hircus* TRB locus contig as previously characterized [32]) (Appendix A).

The MUSCLE program was used to obtain multiple alignments of the gene sequences under analysis [33]. The evolutionary analyses were conducted in MEGA7 [34]. To reconstruct the phylogenetic tree we used the neighbor-joining (NJ) method [35] while the evolutionary distances were computed using the p-distance method [36].

### 2.4. 5′ Rapid Amplification of cDNA Ends (RACE) PCR

Blood samples were provided by Zoomarine Italia S.p.A. (Rome, Italy) and were collected from two unrelated dolphins, a male (Marco) and a female (Leah), identified with letters M and L, respectively.

Total RNA was isolated from peripheral blood leukocytes (PBL) using the Trizol method according to the manufacturer’s protocol (Invitrogen, Carlsbad, CA, USA). About 5 μg of RNA was reverse transcribed with Superscript II (Invitrogen, Carlsbad, CA, USA) by using a specific primer, TCB1L1 (5′ GCTGGGGTCCTCCTTGTC 3′). After linking a poly-C tail at the 5′end of the cDNAss, the cDNAds was performed with Platinum Taq Polymerase (Invitrogen) by using a specific primer, TCB1L2 (5′ TTCCCGTTCACCCACCAG 3′) as lower primer and an anchor oligonucleotide as upper primer (AAP) provided from the supplier (Invitrogen). PCR conditions were the following: one cycle at 94 °C for 1 min; 35 cycles at 94 °C for 30 s, 58 °C for 45 s, 72 °C for 1 min; a final cycle of 30 min at 72 °C. The products were then amplified in a subsequent nested PCR experiment by using a specific lower primer, TCB1L3 (5′ GTGGTGACGGGGTAGAAG 3′) and Abridged Universal Amplification Primer (AUAP) oligonucleotide as upper primer, provided from the supplier (Invitrogen). Nested PCR conditions were the following: one cycle at 94 °C for 1 min; 30 cycles at 94 °C for 30 s, 58 °C for 35 s, 72 °C for 30 s; a final cycle of 30 min at 72 °C. All the specific primers, TCB1L1, TCB1L2, and TCB1L3 were designed on the sequence of the first exon of the TRBC gene. The rapid amplification of cDNA ends (RACE) products were then gel-purified and cloned using the StrataClone PCR Cloning Kit (Stratagene). Random selected positive clones for each cloning were sequenced by a commercial service. cDNA sequence data were processed and analyzed using the BLAST program (http://www.blast.ncbi.nlm.nih.gov/Blast.cgi, accessed on 10 February 2021), Clustal Omega alignment tool (http://www.ebi.ac.uk, accessed on 10 February 2021), and IMGT tools (IMGT/V-QUEST [37,38] with integrated IMGT/JunctionAnalysis [39,40] and the IMGT unique numbering for the V-domain [17,23]). All cDNA clones were registered in the EMBL database with the accession numbers from HG764428 to HG764459.

### 2.5. Crystal Structure Sampling Via Folding Recognition and Multiple Sequence Alignments

The folding recognition methods implemented in pGenThreader and i-Tasser were used for highlighting TR alpha and TR beta homologous protein-crystallized structures. With this aim, the deduced amino acid sequences of the *T. truncatus* TRA (TRAV20-S2*01, 5RAL27 clone, GenBank: LN610732.1; TRAV18-1*02-J20, 5RAL11 clone, GenBank: LN610716.1) and TRB (TRBV30-D2-J2-2, L5RBL27 clone, clone 5RBL27 in GenBank: HG764454.1; TRBV30-D2-J3-3, L5RBL13 clone, clone 5RBL13 in GenBank: HG764440.1) cDNAs were used as query sequences for running pGenThreader (http://bioinf.cs.ucl.ac.uk/psipred/, accessed on 10 February 2021) and i-Tasser (https://zhanglab.ccmb.med.umich.edu/I-TASSER/, accessed on 10 February 2021) to screen the PDB, searching for the most similar deposited crystallized structures [41,42,43,44,45]. SPDBV [46] was used for building a 3D all-atom model of all the investigated *T. truncatus* TR alpha or TR beta chains by using the human TR alpha or TR beta available in the human crystallized TR in complex with major histocompatibility class I (MH1) protein made of an I-ALPHA chain noncovalently associated with beta-2-microglobulin (B2M) [23], available under the pdb code “3hg1.pdb”.

The human major histocompatibility class I (MH1 I-ALPHA) protein and beta-2 microglobulin (B2M) sequences were used as query sequences for identifying their counterparts in *T. truncatus*, through BLAST searches. The identified protein sequences were modeled by using the structures of the human MH1 I-ALPHA and B2M available under the pdb code “3hg1.pdb”, used as a protein template, by SPDB according to our validated protocols [44,45,47]. All the generated 3D all-atom models were energetically minimized by using the Yasara Minimization server [45,47,48].

### 2.6. 3D Modeling of T. truncatus TR Alpha and TR Beta in Complex with T. truncatus MH1I-ALPHA and B2M

The proposed 3D comparative protein complex consisting of *T. truncatus* TR alpha and TR beta in complex with *T. truncatus* MH1 I-ALPHA and B2M was obtained by superimposing the above-cited single chains domains (*T. truncatus* TR alpha, TR beta, MH1 I-ALPHA, and B2M) on the corresponding 3D atomic coordinates of each corresponding chain within the crystallized human TR in complex with human MH1 I-ALPHA and B2M, available under the PDB code 3hg1.pdb [49], by PyMOL as previously described [45,47].

It should be noted that CANCER/MART-1 decapeptide was removed from the 3hg1 before starting the comparative modeling analysis. Superimposition operations were performed through the “super” command implemented in PyMOL, starting from the structural alignment of the analyzed backbones. The “super” command allows aligning the selected proteins under investigation for performing a comparative structural analysis, due to its ability in providing a sequence-independent structure-based pairwise alignment. Notably, the “super” command is more robust than the “align” command because it also successfully performs superimposition of proteins with a lower sequence similarity. Then, it was possible to model/relax missing/buried residues located at the protein–protein interface, solving clashes and putative breaks in the backbone [43,45,47]. All the generated 3D all-atom models were energetically minimized by using the Yasara Minimization server [45,47,48]. The obtained final models were examined in VMD, PyMOL, and SPDBV by visual inspection searching for putative unsolved clashes [44,45,47]. Protein–protein binding regions were highlighted by selecting residues within 4 Å at the protein–protein interface, in the superimposed structures.

The FoldX AnalyseComplex assay was performed to determine the interaction energy between the four generated *T. truncatus* TR alpha and TR beta models and between each of the four combined TR alpha/beta in complex with the *T. truncatus* MH1 I-ALPHA and B2M, but also for determining the interaction energy between the human counterparts of the crystallized 3hg1.pdb as a reference system, a validation strategy, and for comparative purposes.

The way the FoldX AnalyseComplex operates is by unfolding the selected targets and determining the stability of the remaining molecules and then subtracting the sum of the individual energies from global energy. More negative energies indicate a better binding. Positive energies indicate no binding [50,51].

## 3. Results

### 3.1. Genomic Structure of the Tursiops truncatus TRB Locus

We employed the latest version of the whole genome assembly (mTurTru1.mat.Y) of the bottlenose dolphin (*T. truncatus*) submitted by the Vertebrate Genomes Project, to NCBI (BioProject ID: PRJNA625792) to identify the TRB locus in this species. We retrieved a sequence approximately 278 kilobase (kb) in length (gaps included), comprising the MOXD2 and the EPHB6 genes that flank the 5′ and 3′ ends, respectively, of all mammalian TRB loci studied to date.

A standard BLAST search of the genomic sequence was then performed by using human [21,23] and artiodactyl sequences i.e., *Sus scrofa*, *Camelus dromedarius*, and *Ovis aries* [24,25,26,27] to identify and annotate all dolphin TRB genes.

The sequence analysis showed a conserved general structural organization of the dolphin TRB locus with a library of TRBV genes positioned at the 5′ end of the D-J-C clusters, followed by a single TRBV gene located at the 3′ end in an inverted transcriptional orientation (Figure 1A). Only functional TRBV genes and in-frame pseudogenes are shown in Figure 1B.

Moreover, it revealed that the D-J-C cluster is organized in three D-J-C sets similar to those found in ruminants [27,53], pig [24], and camel species [25,26,54], confirming the tight evolutionary relationship between Cetacea and Artiodactyla. D-J-C cluster 1 contains one TRBD, six TRBJ genes but lacks the TRBC gene, due to a gap in the genomic sequence. Likewise, the second set (D-J-C cluster 3) lacks the TRBD gene and it includes seven TRBJ and one TRBC gene. D-J-C cluster 2 is the only complete set with one TRBD, seven TRBJ, and one TRBC gene (Figure 1A, Appendix A). Approximately 10 kb away from the TRBC2 gene lies the TRBV30 gene, with an inverted transcriptional orientation. The classification, position, and predicted functionality of all TRB genes are reported in Appendix A.

The comparison of the entire dolphin TRB locus sequence with those previously characterized in mammals, allowed us to identify and partially annotate four unrelated TRB genes consisting of a group of trypsin-like serine protease (TRY) genes that are typically interspersed among the TRB genes. Three TRY genes are located downstream of TRBV1, and one (TRY4) is located upstream of the D-J-C region (Figure 1A, Appendix A). The gene-predicted functionality (only the TRY3 gene is functional) is reported in Appendix A, together with the position of all TRY genes. The classification, position, and predicted functionality of the MOXD2 and EPHB6 genes, which delimit the TRB locus, are also reported.

### 3.2. Description of the D-J-C Clusters Region

The structure of the dolphin D-J-C clusters region is similar to that of other artiodactyl species [24,25,26,27,32,53,54]. The TRBD, TRBJ, and TRBC genes are distributed within three in tandem D-J-C clusters located at the 3′ end of the TRB locus, with the numbers 1, 3, and 2 corresponding to their positions from 5′ to 3′ (Figure 1A, Appendix A). The name D-J-C cluster 3 was attributed to the central cluster as in sheep, goat, dromedary, and pig [24,25,26,27,32,54].

The nucleotides and deduced amino acid sequences of all the TRBJ genes identified in the region are reported in Figure 2A. Due to the lack of the TRBC1, the TRBJ1 genes were assessed by comparison with the human and artiodactyl corresponding gene sequences. All TRBJ genes are typically 40–53 bp in length and are predicted to be functional, except for TRBJ1-5 and TRBJ3-6 pseudogenes for a STOP-CODON in the coding region (Figure 2A, Appendix A) and TRBJ1-2, TRBJ1-4, and TRBJ3-2 classified as an open reading frame (ORF). Each TRBJ is flanked at the 5′ end by a 12 RS (NONAMER—12 bp SPACER—HEPTAMER) and at the 3′ end by a donor splice site. All the RSs are well conserved compared to the consensus. Figure 2B shows the nucleotide and deduced amino acid sequences of the only two TRBD genes. They are composed of G-rich stretches of 13 bp (TRBD1) and 15 bp (TRBD2), respectively, that can be productively read through their three coding phases and encode 1–3 glycine residues, depending on the phase. The 5′ and 3′ sides of the coding region are flanked by the RSs that are well conserved compared to the consensus.

A protein of 177 amino acids is encoded by the two dolphin available TRBC genes composed of four exons and three introns. Ten nucleotides are different and this results in three amino acid (AA) changes, two in the extracellular and one in cytoplasmic regions (Figure 2C). The C-domain encoded by EX1 is 129 AA long as in humans and pigs because of the identical length of the FG loop compared to other artiodactyl species. Based on the IMGT unique domain for C-DOMAIN [55], the C region is composed of a connecting region (CO) of 21 AA (encoded by EX2 and the 5′ part of EX3) with a cysteine in the interchain disulfide binding, a TM of 21 AA (encoded by the 3′ part of EX3 and the first codon of EX4) and a cytoplasmic region (CY) of 5 AA (encoded by EX4).

### 3.3. Classification and Phylogenetic Analysis of the Dolphin TRBV Genes

In the retrieved sequence, we annotated 23 TRBV germline genes grouped into 20 distinct subgroups according to the criterion that sequences with a nucleotide identity of more than 75% in the V-region belong to the same subgroup. Only two subgroups have more than one member, TRBV5 and TRBV7 with three and two genes, respectively.

Twelve out of twenty-three TRBV genes (approximately 50%) are predicted to be functional as defined by the IMGT rules (see “Materials and Methods”), and eleven are pseudogenes (Table 1 and Appendix A).

The classification and nomenclature of the dolphin TRBV gene subgroups were established first by the IMGT/V-QUEST tool [37,38], comparing each V-region sequence with the germline human, pig, and sheep TRBV sequence set from the IMGT reference directory. In the Cetartiodactyla superorder, the gene nomenclature based on IMGT standardized rules [23] has been previously well defined [24,25,26,32].

In Figure 1B the TRBV potential functional genes and in-frame pseudogenes are shown. All sequences exhibit the typical framework regions (FR-IMGT) and complementarity-determining regions (CDR-IMGT) as well as four amino acids: cysteine 23 (1st-CYS) in FR1-IMGT, tryptophan 41 (CONSERVED-TRP) in FR2-IMGT, hydrophobic amino acid 89, and cysteine 104 (2nd-CYS) in FR3-IMGT (except for the TRBV6 pseudogene) [52]. Conversely, CDR-IMGT varies in amino acid composition and length, indicated by numbers between brackets, separated by dots.

The phylogenetic analysis was performed comparing all dolphin TRBV functional genes, ORF, and pseudogenes with the available corresponding pig, dromedary, and goat genes whose sequences were chosen by adopting two selection criteria: (1) only potential functional genes and in-frame pseudogenes were included; and (2) only one gene per each of the subgroups was selected.

An unrooted phylogenetic tree was made using the NJ method [35] combining the V-REGION nucleotide sequences of all selected TRBV genes in the same alignment. The tree shows that each of the 20 dolphin subgroups forms a monophyletic group, when present, with the corresponding pig, dromedary, and goat genes, this being consistent with the occurrence of distinct subgroups prior to the divergence of the mammalian species. The phylogenetic clustering confirmed the classification of the dolphin TRBV genes derived from the sequence analysis. Therefore, each dolphin TRBV subgroup was classified as orthologous to the corresponding artiodactyl subgroups (Figure 3).

As summarized in Table 1, the total number of the dolphin TRBV genes is considerably lower if compared with the artiodactyl species, with humans and carnivores too, mainly due to the lack of several TRBV subgroups. Particularly, eight (TRBV3, TRBV9, TRBV14, TRBV15, TRBV16, TRBV21, TRBV23, and TRBV25) subgroups were not found in dolphin (dark balls in Figure 3). TRBV13 and ORF TRBV17 genes, present in humans [8], are missing in cetartiodactyls (dolphin, goat, pig, and dromedary) and carnivores (dog) [58] genomes. The lack of the dolphin TRBV18 is shared with pig and dromedary; pig TRB locus lacks TRBV16 and TRBV26, while goat and dog loci lack the TRBV23 subgroups. The TRBV9 subgroup was not found in pig and dog genomes whereas the TRBV14 is absent from the dog locus (Table 1).

### 3.4. Genomic Structure and Evolution of the Dolphin TRBV Locus

Genomic comparison of the dolphin TRB locus with that of the human, dromedarius, pig, and goat, highlighted the uniqueness of the structure of the dolphin TRBV locus. Twenty-three dolphin TRBV genes, grouped into 20 distinct subgroups, lie in a region of approximately 145 kb (Figure 1A). The dot-plot matrix of the dolphin versus human TRB locus confirms the high level of nucleotide identity between TRBV genes as indicated by dots and diagonal lines that correspond with gene location (Figure 4).

The only region where gene duplication took place in *T. truncatus* TRB locus concerns the TRBV5 and TRBV7 genes, which are inserted in a group consisting of duplications of individual TRBV5 (from 5-1 to 5-3), and TRBV7 (from 7-1 to 7-2) pseudogenes, and are interspersed with the TRBV6 pseudogene and with the TRBV8 ORF (Figure 1A and green line in Figure 4). This group of six pseudogenes and a single ORF is found in the matrix showing the dot-plot of dolphin against the dromedarius TRBV cluster (green line in Appendix A) and against the pig TRBV cluster sequences (green line in Appendix A). This group of genes whose sequential order on the genome is conserved in cetaceans, swine and Tylopoda, occupies about 15 kb of the TRB locus in the dolphin genome.

On the contrary, the corresponding group in humans went through duplications up to eight copies for TRBV5 (TRBV5-8) and up to nine copies for TRBV6 and TRBV7 (TRBV6-9, TRBV7-9). It should be noted that the latter are all functional genes, with the exception of a single pseudogene (TRBV7-5). Such duplications occupy a region extending for 200 kb (green rectangles in Figure 3) which is interrupted by a portion of 50 kb (red rectangle in Figure 4) containing the only copy of TRBV9 gene and the duplication of the group consisting of TRBV10, TRBV11 and TRBV12 genes (Figure 4). In *Capra hircus*, the TRB locus [32], in correspondence of the dolphin TRBV5-1, TRBV5-2, and TRBV6 genes, houses a series of repeated duplicative events that lead to the existence of 30 copies for the TRBV5 and 29 for TRBV6, for a total of 59 (29 functional, 26 pseudogenes, and 4 ORF) genes. This region seat of repeated duplicative events extends without interruption for 180 Kb (red rectangle in Appendix A).

Furthermore, in the human-dolphin genome comparative analysis, the region including seven genes (from 12-4 to 18) in humans is deleted in the dolphin (orange rectangle in Figure 4) where TRBV12 and TRBV19 genes are neighbors in the same region (Figure 1A). If compared to the other cetartiodactyls, this deletion (orange rectangle) in dromedary (Appendix A), in pig (Appendix A), and in goat (Appendix A) matrices, is unique and typical of the dolphin TRB locus. The deleted region marked with an orange square in Figure 4 is due to the presence of the interspersed deletions of TRBV23 and TRBV25 dolphin genes in comparison with the human contig. Finally, the dot-plot matrix in Figure 4 shows five lines of similarity indicating the duplications in human of TRY4 (from TRY4 to TRY8) (blue rectangles) and five dots in correspondence of the only functional TRY gene (TRY3) (blue rectangles) present in dolphin (Appendix A).

### 3.5. 5′ RACE PCR Assay

To evaluate the functional competency of the three D-J-C clusters, we performed three 5′ RACE experiments on total RNA isolated from the peripheral blood of two unrelated animals, a male (M) and a female (L), using a TRBC-specific primer. Each RACE product was gel-purified and cloned into the TA-vector and randomly selected positive clones for each cloning were sequenced [9]. A total of 45 diverse clones of different lengths containing rearranged V-(D)-J-C transcripts with a correct open reading frame were obtained. In particular, 22 out of 45 cDNA clones belonged to the male subject (M5RBL series), while the remaining 23 cDNAs were derived from the female subject (L5RBL series) (Figure 5). Five clones of the L5RBL series (L5RBL12, L5RBL13, L5RBL20, L5RBL2, and L5RBL27), and three clones of the M5RBL series (M5RBL15, M5RBL22, and M5RBL30) showed an identical coding sequence for a total of 13 redundant clones.

Each of the remaining 32 cDNA sequences was manually analyzed to identify the TRBV, TRBD, and TRBJ genes through alignment with the germline dolphin TR genes. Figure 5 shows the deduced amino acid sequences of the V-(D)-J-region of all cDNA clones according to IMGT unique numbering for the V-REGION and V-DOMAIN [52]. In the cDNA clones, we identified 7 different TRBV genes belonging to 7 out of 12 different subgroups consisting of predicted functional germline genes. In 16 out of 32 clones, the TRBV gene perfectly matched the corresponding germline sequence. The remaining 16 TRBV sequences showed a nucleotide identity from 97 to 99% with respect to the reference germline gene sequence. We referred to these as new alleles, although we cannot rule out sequencing errors or that duplicated genes have been missed. The TRBV30 subgroup gene represented the most expressed (17 unique different clones, 53%) followed by the TRBV12 (7 clones) and the TRBV4 (4 clones) genes. Only one clone was found for four (TRBV10, TRBV20, TRBV22, and TRBV27) subgroups. Moreover, the functionality of the dolphin TRBV12 gene was confirmed since originated seven productive transcripts despite the presence of a stop-codon in the germline CDR3-IMGT at position 108 (Figure 1B) [52] (IMGT Repertoire, Alignments of alleles, http://www.imgt.org, accessed on 10 February 2021), that was trimmed during rearrangement (Figure 5 and Figure 6). In both subjects analyzed, of the 12 functional genes present in the locus (Figure 1A,B) and described in Table 1, five of them (TRBV1, TRBV8, TRBV19, TRBV24, and TRBV26) were not found expressed in peripheral blood (Figure 5).

### 3.6. Analysis of the V-D-J Rearrangements

A certain difference in the two analyzed individuals was observed in the usage of the TRBV genes in the recombinative events. In fact, all TRBV subgroups were found in the cDNA from the male subject (M5RBL series) while only 3 out 7 TRBV subgroups were identified in the female (L5RBL series) cDNA. It is interesting to note that the TRBV30 gene is almost equally recurrent (9 out of 23 L5RBL series and 8 out of 22 M5RBL series) in the transcripts of the two animals, suggesting that this gene may be involved in the expression of a public TRB repertoire (allele*02 shared in both individuals).

All TRBJ genes within the set of cDNA sequences were assigned unambiguously to the distinct germline genes derived from the reference sequence and only in two cases (TRBJ2-7 and TRBJ3-1) we identified new alleles, which were labeled by consecutive numbers with respect to the germline allele *01 (Figure 5). The TRBJ2 set genes with a frequency of 20/32 clones appear to be preferentially expressed over the TRBJ3 set genes with a frequency of 11/32 clones. The surprising result concerns the almost absence of the use of the TRBJ1-1 gene with a single (L5RBL4) clone that contains it (Figure 5 and Figure 6).

More complex is the determination of the contribution of the TRBD genes involved in the CDR3 formation. For a close inspection, the nucleotide sequences corresponding to the CDR3 have been excised from each cDNA and analyzed in detail (Figure 6). By comparison with the TRBD genomic sequences, the nucleotides located in the CDR3 were considered to belong to a TRBD gene if they constituted a stretch of at least five consecutive nucleotides. We were able to assess the contribution of only the TRBD1 and TRBD2 germline sequences but was not possible to identify the presence of a third TRBD gene because this is missing in the current genomic assembly. In this way, the TRBD was unambiguously identified in 21 out of 32 sequences (62.5%) with the TRBD1 present in 16 and TRBD2 in 5 clones, respectively. The remaining 11 sequences either do not have an identifiable TRBD gene, maybe for the lack of the TRBD3 reference sequence, or it was not possible to distinguish between TRBD1 and TRBD2 genes. However, the absence of a TRBD region could be also interpreted as a direct V-J junction and it is even possible that nucleotide trimming masked the initial participation of the TRBD gene during the rearrangement. The corresponding amino acid sequence of the CDR3 loop deduced from the nucleotide sequences is heterogeneous for amino acid composition and length (Figure 5 and Figure 6). The mean length of the CDR3 loop was approximately the same in the dolphin (mean 12,3 amino acids in a range of 5-16 amino acids) with respect to the human (mean 12,7 amino acids; [59]), pig (mean 12,2 amino acids; [24]), camel (mean 12,8 amino acids; [25]) and goat (mean 12,3 amino acids; [32]).

Finally, although the TRBD gene was not unambiguously recognizable in all of the cDNA clones, the interpretation of these rearrangements revealed that only five clones show the intracluster rearrangements with TRBD2-TRBJ2 (yellow in Figure 6). Intercluster rearrangements represent a substantial portion of the repertoire, with 9 TRBD1–TRBJ2 (light blue in Figure 6) and 7 TRBD1–TRBJ3 (green in Figure 6) rearrangements.

### 3.7. 3D Comparative Modeling of the T. truncatus TR Alpha/Beta Chain in Complex with MH1 I-ALPHA and B2M

To calculate the most likely interactions between the putative alpha/beta pairing in complex with MH1 I-ALPHA and B2M, we analyzed the amino acid sequences of the two types of rearranged TRA cDNA, originated by TRAV20S2*01–J17 (5RAL27) and TRAV18-1*02–J20 (5RAL11) rearrangements, respectively found in the peripheral blood of the animal identified with the letter L in our previous study [9], and of relevant TRB cDNA, originated by TRBV30–D2–J3 (L5RBL13) and TRBV30–D2–J2 (L5RBL27) rearrangements among the nine found in this study in the peripheral blood of the same animal (TRBV30 clones initialed by the initial letter L in Figure 5 and Figure 6).

The 3hg1.pdb crystallized structure consisting of the human TR alpha/beta chains in complex with MH1 I-ALPHA and B2M was identified along with pGenThreader and I-tasser searches and used as a protein template for guiding the 3D comparative modeling analysis (Figure 7). *T. truncatus* TR alpha and TR beta chains were modeled by using as a protein template 3hg1.pdb chain D, the human TR alpha, sharing with the modeled *T. truncatus* TR alpha putatively coded by 5RAL27 or 5RAL11 clonotypes, 46 or 41% of identical amino acids, respectively, or 3hg1.pdb chain E, the human TR beta, sharing with *T. truncatus* TR beta, putatively coded by L5RBL13 or L5RBL27 clonotypes, 59–71% of identical AA residues, respectively, and 100% coverage. All *T. truncatus* TRA and TRB sequences showed 100% coverage with the corresponding human TR alpha or TR beta crystallized chains.

The pairwise sequence-structure alignment between the investigated *T. truncatus* TR alpha and TR beta and the human crystallized counterpart (from 3hg1.pdb) was obtained by using ClustalW [60] implemented in the Jalview package [61]. The obtained sequence-structure pairwise alignment (Appendix A) was reported in the SPDBV alignment panel for guiding/building the 3D comparative model of both TRA and TRB chains, according to validated protocols [41,42,43,44,45].

For studying interactions between *T. truncatus* TR alpha and TR beta with MH1 I-ALPHA in *T. truncatus* we searched for *T. truncatus* MH1 I-ALPHA and we identified the sequence XP_033706382.1 sharing with the human MH1 I-ALPHA (3hg1.pdb, chain A) sequence more than 79% of identical amino acids and 100% coverage. Similarly, we searched for *T. truncatus* best hit B2M counterpart, and the sequence XP_033707513.1 was identified, sharing with the human B2M sequence more than 74% of identical amino acids and 100% coverage. A 3D comparative model of the *T. truncatus* MH1 I-ALPHA and B2M was built by using as a protein template the corresponding counterparts taken from 3hg1.pdb starting from the pairwise sequence-structure alignment built by CulstalW, as proposed for TR alpha and TR beta (Figure 7 and Appendix A). The root mean square deviation (RMSD) between the coordinates of the built 3D comparative models and the crystallized 3hg1.pdb ranged between 0.15 and 0.5 Å.

### 3.8. Binding Energy Calculations at the Protein Interface between TR Alpha and TR Beta Chains: Computational Analyses Predict the Pairing of the TRAV20-J17 (or TRAV18-J20) and TRBV30-D2-J3-3 Variable Domains

The interaction energies calculated between the TR alpha and TR beta and between the TR alpha/beta and MH1 I-ALPHA resulted in a negative value (Table 2), confirming that there might be a binding interaction in all the investigated cases. This result is encouraging, also due to the strategy validated by obtaining negative binding energies for the interactions between the human TR alpha and TR beta chains and the MH1 I-ALPHA chain within the crystallized complex available under the PDB code “3hg1.pdb”.

Notably, the strongest binding interactions between the investigated *T. truncatus* TR alpha and TR beta 3D models is observed in the L5RBL13_5RAL27 clonotypes containing protein complex and in the L5RBL13_5RAL11 clonotypes containing protein complex (in terms of interaction energies calculated by FoldX Analyse complex assay) (Table 2, Figure 8A,B), whereas the strongest interaction between TR alpha/beta and MH1 I-ALPHA is observed in the L5RBL27_5RAL27 clonotypes containing protein complex (Table 2).

While interaction energies estimated at the protein interface within the modeled *T. truncatus* TR alpha and TR beta appear weaker than their interacting counterparts in the crystallized structure (3hg1.pdb), interaction energies calculated at the protein interface between the modeled *T. truncatus* TR alpha/beta and MH1 I-ALPHA appear stronger than their counterpart in the human 3hg1.pdb in all the cases (Table 2).

Interacting amino acids located in the CDR-IMGT of the variable alpha (V-ALPHA) and beta (V-BETA) domains of the TRA/TRB chains are shown in Figure 8A,B. It is interesting to note the recurring position in the DE turn [18] of the FR3-IMGT regarding the L5RBL13 clonotype (red arrow indicating E84 in Figure 8B), the 5RAL27 clonotype (red arrow indicating K 85 in Figure 8A), and the 5RAL11 clonotype (red arrow indicating T 82 in Figure 8B).

For a complete picture, according to comparative modelization, a detailed atom pair contact sites analysis in the human 3hg1.pdb based on the IMGT unique numbering is reported in Appendix A, Figure 9 and Figure 10. “IMGT pMH contact sites” [63,64] precisely identify the contacts between the amino acids of a presented peptide and those of the floor and helix walls of the MH groove, in 3D structures of pMH and TR/pMH complexes. Eleven standard “IMGT pMH contact sites” were defined (C1–C11). They correspond to a theoretical maximum length of 11 AA in the groove [63,64]. The peptide binding mode to MH1 is characterized by the N-terminal and C-terminal peptide ends docked deeply with the C1 and C11 contact sites (red and pink, respectively in the IMGT Collier de Perles, Figure 10). For a peptide of 10 AA, one “IMGT pMH contact site” is absent (C2), for a peptide of 9 AA, two “IMGT pMH contact sites” are absent (C2 and C7), whereas, for a peptide of 8 AA, three pMH contact sites are absent (C2, C7, and C8). The characterization of the “IMGT pMH contact sites” based on contact analysis has superseded the previous identification of “pockets” in the MH groove [63,64].

Considering the features of contacts between the *Homo sapiens* G domains and the TR V domains displayed in lines Hs2 of the alignments of the G-ALPHA1 (Figure 9A) and G-ALPHA2 (Figure 9B) domains of *T. truncatus* MH1 I-ALPHA with *Homo sapiens* MH1 I-ALPHA HLA-A*0201, the probability that the same positions are involved in atom pair contacts in dolphins are very high (in Figure 9, in yellow the amino acids in dolphin compared to 3hg1).

## 4. Discussion

In the superorder of Cetartiodactyla, bottlenose dolphin (*T. truncatus*) and the other cetaceans represent the most successful mammalian colonization of the aquatic environment and have undergone a radical transformation from the original mammalian bodyplan. In a previous work [9] we reported in dolphin genomic and expression studies on TR gamma (TRG) and alpha/delta (TRA/TRD) loci. In this study, we report in dolphin an extensive analysis of the genomic organization, evolution, and expression of the TRB genes and the 3D comparative modeling of the TR alpha/beta heterodimer in complex with MH1 I-ALPHA and B2M.

According to comparative analyses with humans and Artiodactyla, i.e., dromedarius [25,26], pig [24], and goat [32], the organization of the dolphin TRB locus is similar to that of the other artiodactyl species, with three in tandem D-J-C clusters located at its 3′ end. In its genomic structure, the dolphin TRB locus is the smallest in length (gaps included), comprising the MOXD2 and the EPHB6 genes that flank the 5′ and 3′ ends, respectively. The overall length of 276 Kb corresponds to less than half of that of *Homo sapiens* (620 Kb), to about half of that of *Capra hircus* (558 Kb) [32], to a little more than half of that of *Sus scrofa* (407 Kb) [24,69], and it is slightly smaller than that of *Camelus dromedarius* (302 Kb) [54]. The uniqueness of the dolphin TRB locus, when compared to the other artiodactyls and to humans, is given both by the almost complete absence of duplications in the TRBV region and by the presence of deletions that drastically reduce the number of the variable genes (see orange rectangles in Figure 4).

An in-depth analysis of the genomic structure of the TRBV genes highlights a ratio of 1/1 of functional genes to pseudogenes against a ratio of about 2/1 in the other examined artiodactyls (*Capra hircus* and *Sus scrofa*) and in carnivores (*Canis lupus familiaris*). A 3/1 ratio is present in *Camelus dromedarius* (Table 1). The phylogenetic analysis shows that each of the 20 dolphin TRBV subgroups forms a monophyletic group, when present, with the corresponding pig, dromedary, and goat genes. This result is consistent with the occurrence of distinct subgroups prior to the divergence of the mammalian species. The phylogenetic clustering (Figure 3) confirmed the classification of the dolphin TRBV genes derived from the sequence analysis.

In the analysis of the TRB V-D-J rearrangements, intercluster rearrangements were observed to constitute a substantial portion of the repertoire, with 10 TRBD1–TRBJ2 and 6 TRBD1–TRBJ3. The fact that the 5′ RACE assay resulted from a primer designed on a region shared by all the constant genes leads us to assume that the 19 clones containing the TRBJ2 set genes and the 12 clones containing the TRBJ3 set genes have the TRBJ gene linked to the corresponding TRBC2 and TRBC3, respectively (Figure 6). Moreover, since the TRBC3 gene is located upstream from the TRBJ2 cluster in the germline DNA (Figure 1A and Appendix A), we cannot rule out that TRBJ2 can be joined to TRBC3 as a product of a trans-splicing between a transcript with TRBJ2-TRBC2 genes and a transcript containing the TRBC3 gene [70].

The expression data highlight the preferential usage of TRBV30 in both the analyzed subjects. Out of a total of 45 cDNA clones obtained from the peripheral blood of the two subjects, TRBV30 is expressed with a percentage of 66.6% (30/45 clones). The redundancy of the clones (from +1 to +3) of the series containing the TRBV30, is highlighted in Figure 5. The preferential usage of the TRBV30 gene, which lies at the 3′ end of the TRBC2 gene in an inverted transcriptional orientation in comparison with other artiodactyls represents a condition of diversity in dolphins. In fact, considering the total number of cDNA clones from peripheral lymphoid organs (spleen and blood), against 30/45 clones in dolphin blood, 2/27 in goat blood [32], 1/26 in pig spleen [24], 0/35 in camel spleen [25], and 0/36 in dog blood [58] are reported in the literature.

The fact that both the unrelated subjects show a biased usage of the TRBV30 could be explained by the sharing by the two subjects of the same aquatic environment. Then the presence of common antigens may have stimulated T cells with a particular type of beta chain to expand, suggesting the existence of a basic “public” repertoire of a given alpha/beta TR [71]. Therefore we used the rearranged TRA [9] and TRB cDNA clones from the peripheral blood of one of the two analyzed subjects (Leah in this work), to build a 3D comparative modeling of the *T. truncatus* TR alpha/beta in complex MH1 I-ALPHA and B2M. The four possible 3D structures (Figure 7b–e) of TRAV20S2*01–J17 (5RAL27) and TRAV18-1*02–J20 (5RAL11) in combination with TRBV30–D2–J3 (L5RBL13) and TRBV30–D2–J2 (L5RBL27) clonotypes (Figure 7) retain a very high similarity with the 3hg1.pdb structure of the human TR alpha/beta chains in complex with MH1 I-ALPHA and B2M.

The negative value resulted from the interaction energies calculated between the TR alpha and TR beta and between the TR alpha/beta and MH1 I-ALPHA (Table 2), allowed the identification of the resulting interacting amino acids located in the CDR-IMGT at a distance of length < 4.00 (Å). In particular, in the L5RBL13 clonotype, Tyr 55 is located on the edge of CDR2 (Figure 8A,B), and in the 5RAL27 clonotype, Tyr 57 on CDR2 (Figure 8A) correlate well with recent literature [72,73]. Piepenbrink et al., in their article, reported that TR and MH proteins were refolded from bacterially expressed inclusion bodies. In this paper, results of the interaction free energy between two amino acid sidechains obtained in a double-mutant cycle pointed out how the engagement of Tyr at the center of the interface by sidechains of CDR TRA and TRB, contributes, depending on its location, significantly to TR affinity [72]. Nevertheless, our results show that interaction energies calculated at the protein interface between the modeled *T. truncatus* TR alpha/beta and MH1 I-ALPHA appear stronger than their counterpart in the human 3hg1.pdb in all the examined cases (Table 2).

## 5. Conclusions

The results of this research confirmed the peculiarity of the genomic organization of the TR loci already identified in previous studies in *T. truncatus*. In the organism of marine mammals, there seems to be an evolutionary pressure that is responsible for reducing the length of the TR loci, in terms of kilobases. This is highlighted thanks to the comparative analyses with other artiodactyls and with humans for both the gamma locus (TRG) in the previous study and the beta (TRB) locus in the present study. The reduction in length of the locus is accompanied by a reduction in the content of the variable genes that are primarily responsible for antigen recognition. Therefore, in addition to having identified the peculiar genomic features and expression of dolphin TRB locus, we have built for the first time in a marine mammal the 3D human/dolphin comparative modeling of the TR alpha/beta in complex with major histocompatibility class I (MH1) protein and beta-2 microglobulin (B2M). Nevertheless, our results of the human/dolphin modelization, integrated with the atom pair contact sites analysis between the human MH1 grove (G) domains and the TR V domains, confirm conservation of the structure of the dolphin TR/pMH.

## Figures and Tables

**Figure 1 genes-12-00571-f001:**
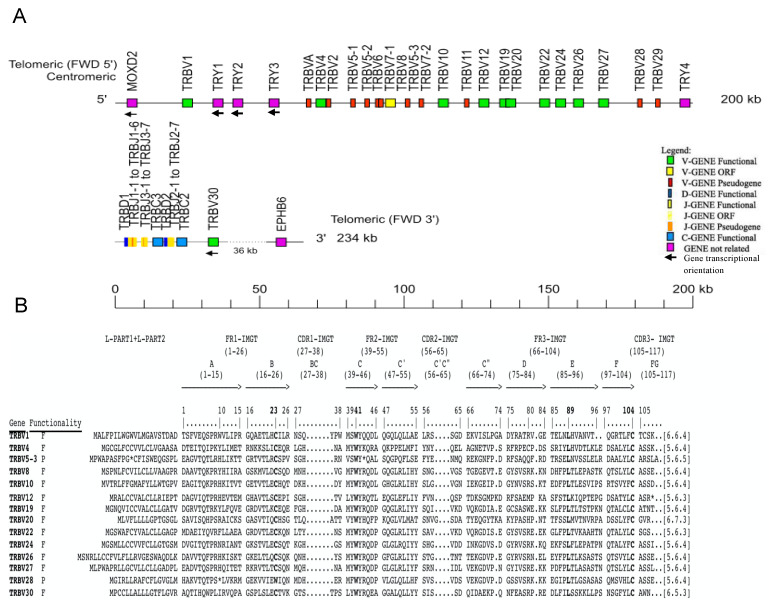
(**A**) Schematic representation of the genomic organization of the bottlenose dolphin (*Tursiops truncatus*) T cell receptor beta (TRB) locus deduced from the genome assembly mTurTru1.mat.Y (CM022282.1). The diagram shows the positions of all the related and unrelated TRB genes according to IMGT nomenclature and locus representation [22]. The boxes representing the genes are not to scale. The exons are not shown. The arrows indicate the transcriptional orientation of the MOXD2, trypsin-like serine protease (TRY) 1, 2, 3, and TRBV30 genes. (**B**) The IMGT Protein Display of the dolphin TRBV genes. Only functional genes and in-frame pseudogenes are shown. The description of the strands and loops and of the framework regions (FR-IMGT) and complementarity-determining region (CDR-IMGT) is according to the IMGT unique numbering for V-REGION [52]. The five conserved amino acids (AA) of the V-DOMAIN (1st-CYS 23, except for the TRBV28 pseudogene, CONSERVED-TRP 41, hydrophobic AA 89, 2nd-CYS 104, and J-PHE 118) are indicated in bold. The amino acid length of the CDR-IMGT AA is also indicated in square brackets.

**Figure 2 genes-12-00571-f002:**
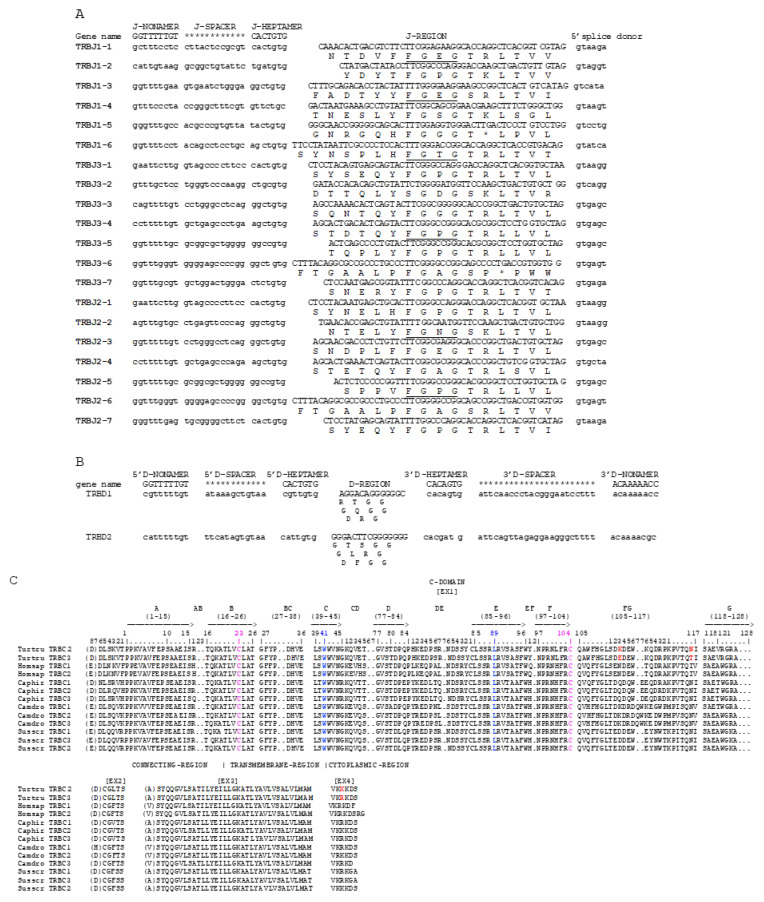
Nucleotide and deduced amino acid sequences of the dolphin TRBD (**A**), TRBJ (**B**), and TRBC (**C**) genes. The numbering adopted for the gene classification is reported on the left of each gene. The consensus sequence of the heptamer and nonamer is provided at the top of the figure and is underlined. In (**A**), the inferred amino acid sequence of the TRBD genes in the three coding frames is reported. In (**B**), the donor splice site for each TRBJ is shown. The canonical FGXG amino acid motifs are underlined. In (**C**), IMGT Protein display of the dolphin TRBC genes as derived from the alignment by Clustal W with the human, sheep, dromedary, and pig TRBC amino acid sequences is shown. The strands and loops are according to the IMGT unique numbering for the C-DOMAIN [55].

**Figure 3 genes-12-00571-f003:**
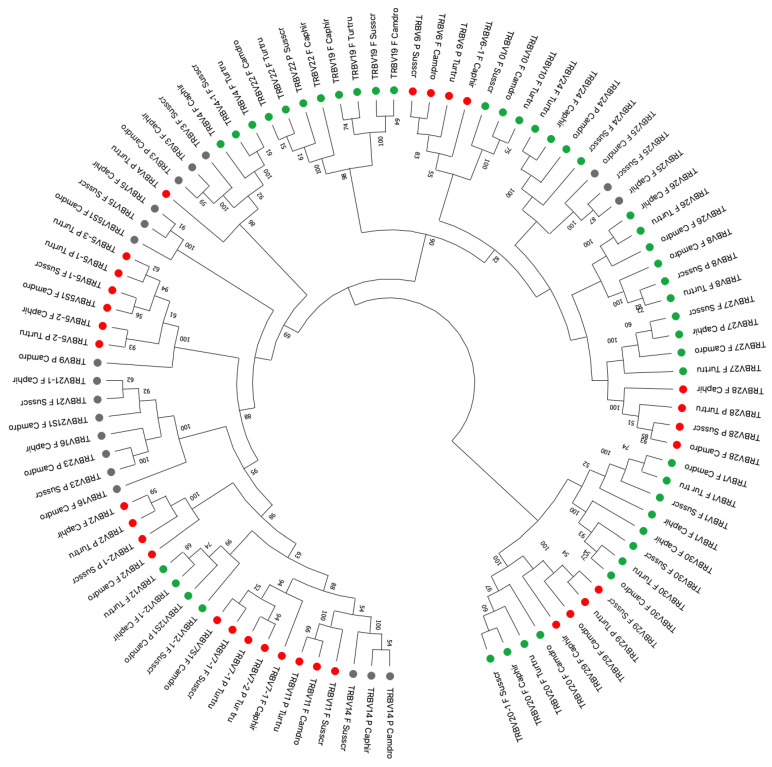
The neighbor-joining (NJ) tree inferred from the dolphin (*Tursiops truncatus*), goat (*Capra hircus*), pig (*Sus scrofa*), and dromedary (*Camelus dromedarius*) TRBV gene sequences. The evolutionary analysis was conducted in MEGAX [34]. The optimal tree with the sum of branch length = 16.02748379 is shown. The percentage of replicate trees in which the associated taxa clustered together in the bootstrap test (100 replicates) is shown next to the branches [56]. The tree is drawn to scale with branch lengths in the same units as those of the evolutionary distances used to infer phylogenetic trees. The evolutionary distances were computed using the Maximum Composite Likelihood method [57] and are in the units of the number of base substitutions per site. This analysis involved 95 nucleotide sequences (Appendix A). All ambiguous positions were removed for each sequence pair (pairwise deletion option). There were a total of 473 positions in the final dataset. The different colors highlight the distribution of the phylogenetic groups (green balls for dolphin functional genes; red balls for dolphin pseudogenes; dark balls for missing genes in dolphin). The dolphin TRBV subgroup classification is performed according to the clustering with the orthologous mammalian TRBV subgroups. The gene functionality according to IMGT rules (F: functional, ORF: open reading frame, P: pseudogene) is indicated. The IMGT 6-letter for species (Turtru, Camdro, Susscr, and Caphir) standardized abbreviation for taxon is used.

**Figure 4 genes-12-00571-f004:**
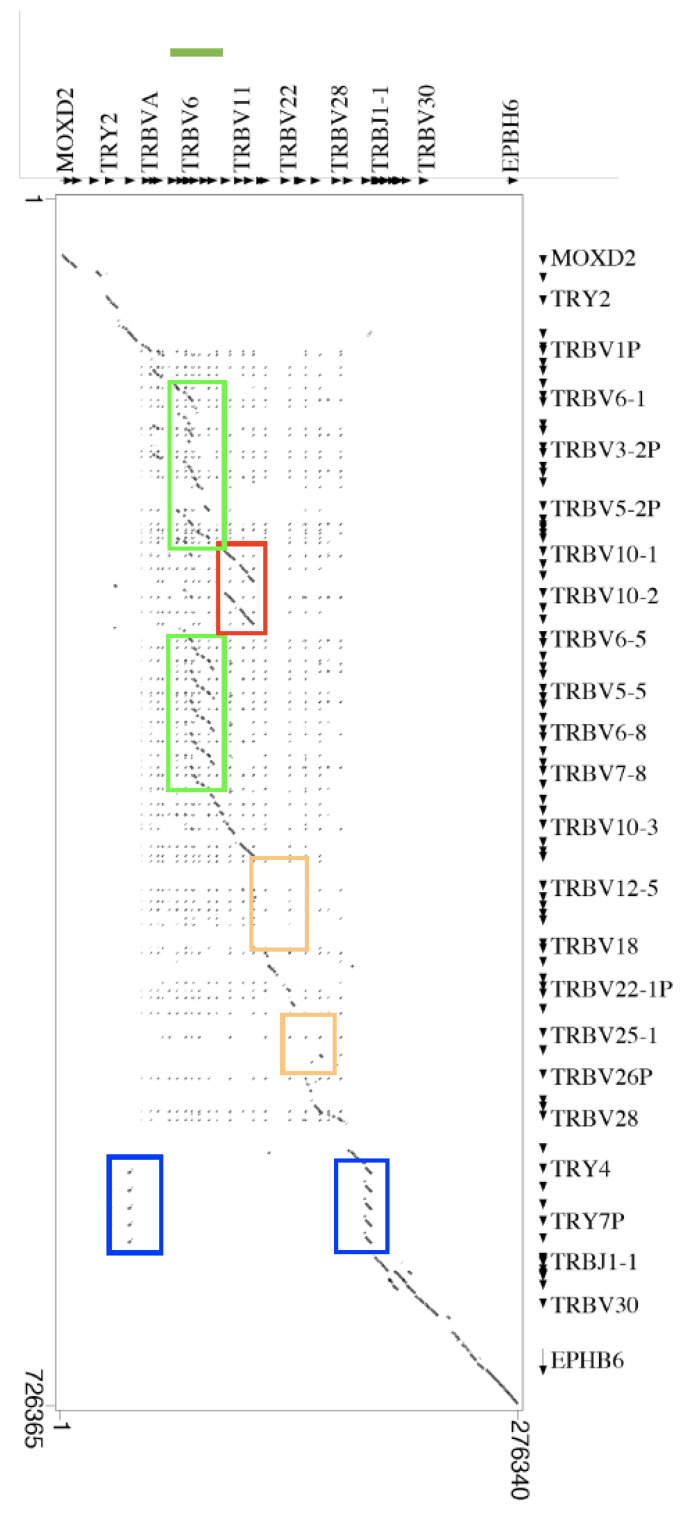
Dot-plot matrix of dolphin/human TRB sequence genomic comparison. The transcriptional orientation of each gene is indicated by arrowheads. Colored rectangles (light green, red, and blue) enclose TRBV duplicated regions in humans and (orange) deleted TRBV regions in dolphins as referred to in the text. The green line indicates the only region where gene duplication took place in *T. truncatus* TRB locus.

**Figure 5 genes-12-00571-f005:**
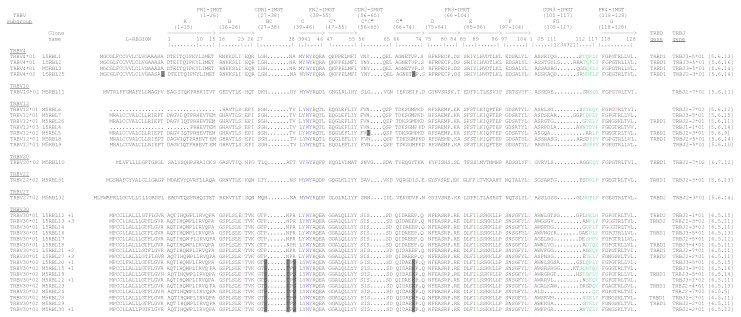
Protein display of the cDNA clones. The TRBV and TRBJ genes, named according to the criteria specified in the text, are listed, respectively, at the left and the right of the figure. Leader region (L-REGION), complementarity-determining regions (CDR-IMGT), and framework regions (FR-IMGT) are also indicated according to the IMGT unique numbering for the V-DOMAIN [52]. The five conserved amino acids of the V-DOMAIN (1st-CYS 23, CONSERVED-TRP 41, hydrophobic AA 89, 2nd-CYS 104, and J-PHE 118) are colored (IMGT color menu). The TRBV allele amino acid changes, if any, are boxed. The name of the clones is also reported.

**Figure 6 genes-12-00571-f006:**
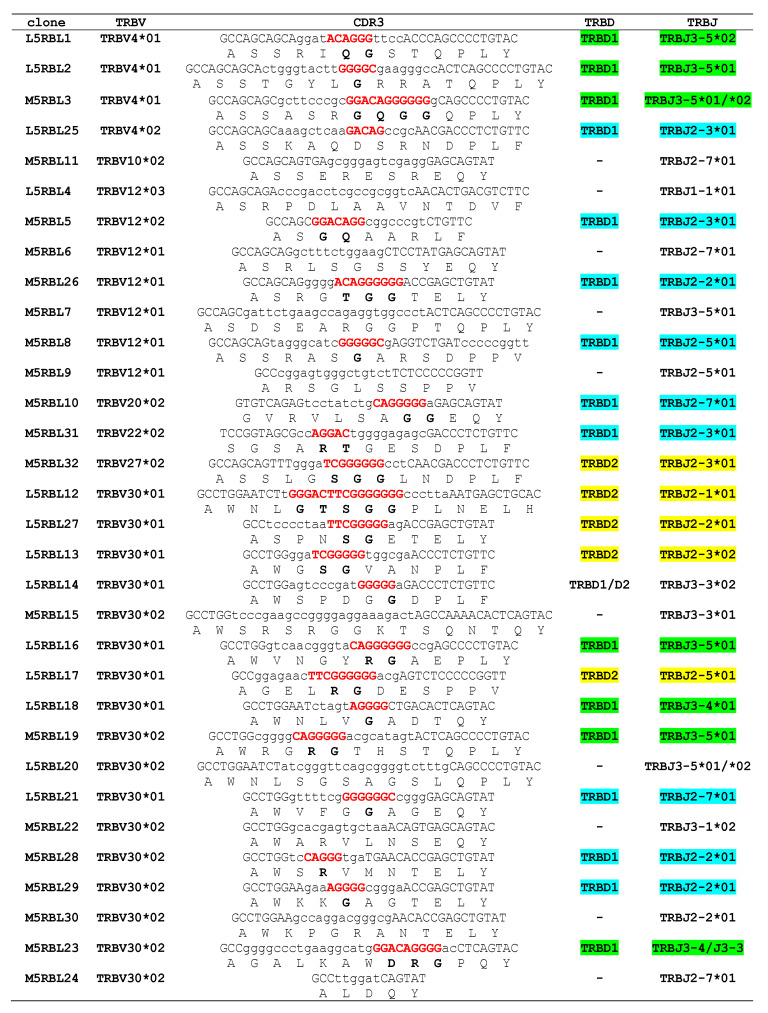
CDR3-IMGT nucleotide and predicted amino acid sequences retrieved from the TRB cDNA clones. CDR3-IMGT sequences are shown from codon 105 (the codon after the 2nd-CYS 104 of the V-REGION) to codon 117 (the codon before J-PHE 118 of the J-REGION) according to the unique numbering [52]. Nucleotides of the 3′V-REGION and of the 5′J-REGION are indicated in uppercase letters. The CDR3-IMGT nucleotide/amino acid sequence, and the classification of the TRBV and TRBJ genes of each clone are also listed. The sequences considered to present recognizable coding regions of the TRBD genes are indicated in red uppercase letters. The amino acids belonging to a TRBD region are in bold. Nucleotides that cannot be attributed to any V, D, or J-region (N-nucleotides), are indicated in lower cases on the left and on the right sides of the TRBD regions. The name of each clone is also reported. Colored clones are referred to in the text.

**Figure 7 genes-12-00571-f007:**
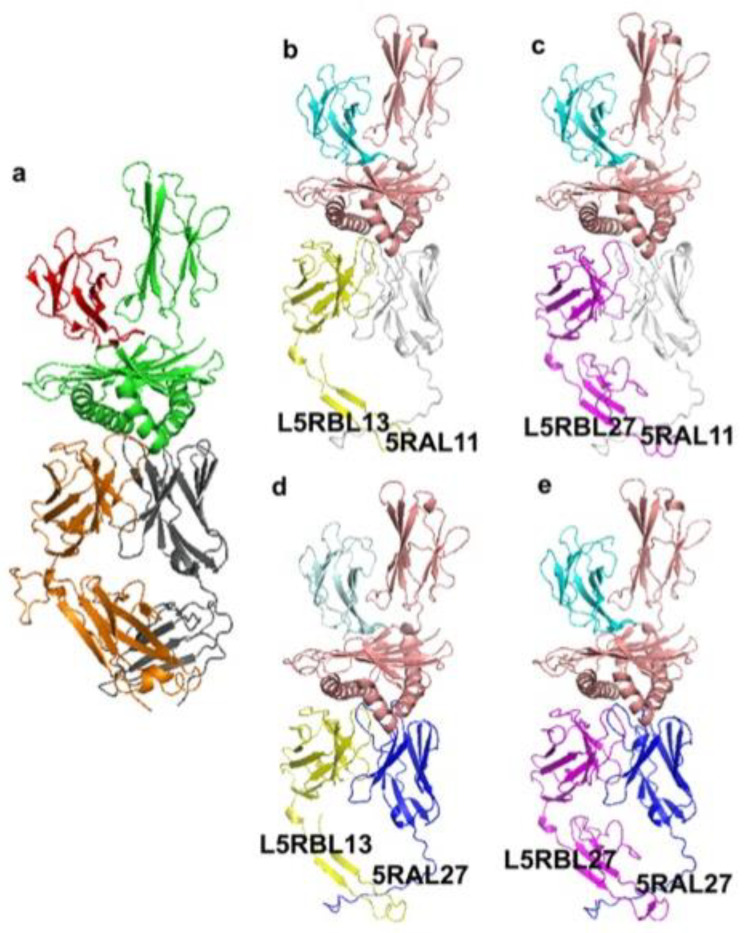
*T. truncatus* TR alpha and TR beta in complex with MH1 I-ALPHA and B2M. (**a**). Human TR alpha and TR beta are reported in grey (TR alpha) and orange (TR beta) cartoon, respectively, in complex with MH1 I-ALPHA (green cartoon) and B2M (red cartoon). (**b**–**e**). Different complex poses showing the investigated combination of *T. truncatus* TR alpha (5RAL11 clonotype, white cartoon; 5RAL27 clonotype, blue cartoon) and TR beta (L5RBL13 clonotype, yellow cartoon; L5RBL27 clonotype, blue cartoon) in complex with MH1 I-ALPHA (salmon cartoon) and of B2M (cyan cartoon).

**Figure 8 genes-12-00571-f008:**
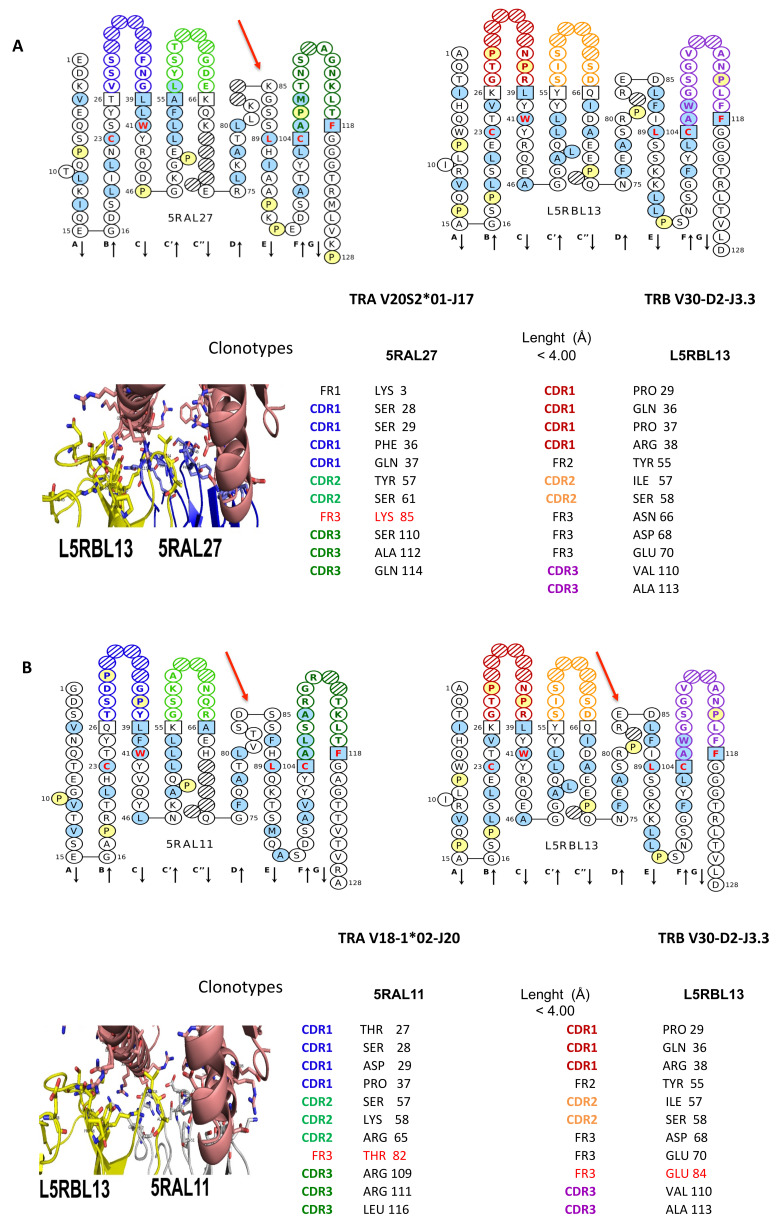
Computationally inferred interaction between 5RAL27 V-alpha domain (TRAV20S2*01-J17) and 5RBL13 V-beta domain (TRB V30-D2-J3.3), (**A**) and between 5RAL11 V-alpha domain (TRA V18-1*02-J20) and L5RBL13 V-beta domain (TRB V30-D2-J3-3). (**B**) cDNA clonotypes. IMGT Collier de Perles of 5RAL27/5RBL13 and 5RAL11/5RBL13 clones are shown [55,62]. In 5RAL11 and 5RAL27 V-alpha domain CDR-IMGT is blue-green-green; in L5RBL13 V-beta domain CDR-IMGT is red-orange-purple. Interacting residues at the interface of the alpha/beta TR 3D models with MH1 I-ALPHA in *T. truncatus* are reported in stick representation (lower left). The protein complex interface was computed by the pGenThreader (http://bioinf.cs.ucl.ac.uk/psipred/, accessed on 10 February 2021) and i-Tasser (https://zhanglab.ccmb.med.umich.edu/I-TASSER/, accessed on 10 February 2021).

**Figure 9 genes-12-00571-f009:**
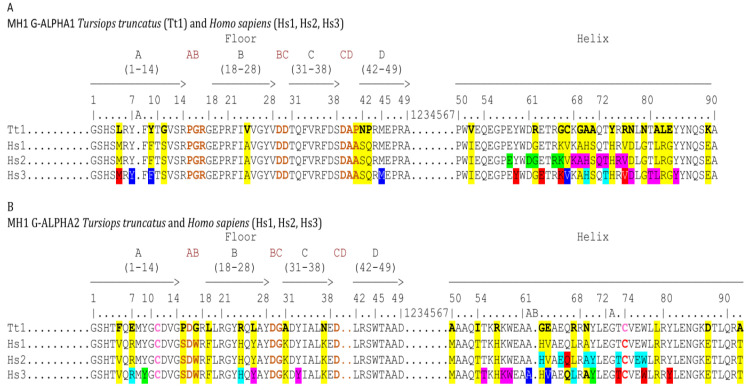
Alignments of the G-ALPHA1 (**A**) and G-ALPHA2 (**B**) domains of *Tursiops truncatus* MH1 I-ALPHA (acc. number: XP_033706382.1) with *Homo sapiens* MH1 I-ALPHA HLA-A*0201. Alignments are according to the IMGT unique numbering for the G domain [65]. Amino acid differences between species are highlighted in yellow (and further in bold in *Tursiops truncatus* lines Tt1). The contacts between the Homsap G domains and the TR V domains are displayed in lines Hs2: in (**A**), G-ALPHA1 contacts with V-ALPHA in green, and V-BETA in pink, in (**B**) G-ALPHA2 contacts with V-ALPHA in blue and V-BETA in red (arbitrary colors). The IMGT pMH contact sites [63,64,66,67,68] between the Homsap G domains and the peptide are displayed in lines Hs3 (colors evoking the contact sites). Positions 61A, 61B, and 72A are characteristic of the G-ALPHA2 and are not reported in G-ALPHA1. Hs1, Hs2 and Hs3 are from 3hg1 in IMGT/3Dstructure-DB, http://www.imgt.org (accessed on 10 February 2021).

**Figure 10 genes-12-00571-f010:**
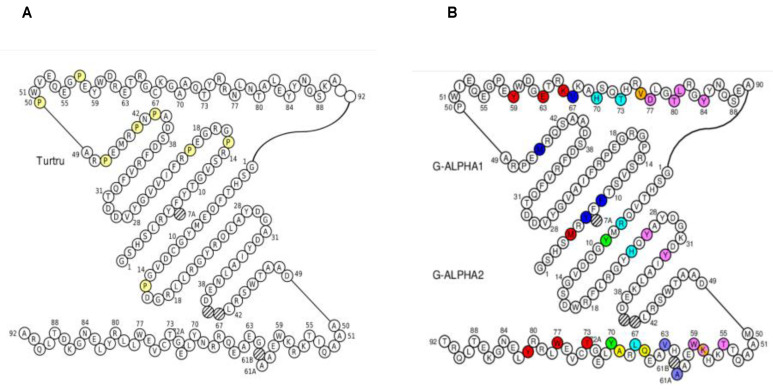
IMGT Collier de Perles of the G-ALPHA1 and G-ALPHA2 of the MH1 I-ALPHA. (**A**), IMGT Collier de Perles of *Tursiops Truncatus*, obtained using the IMGT/Collier-de-Perles tool [68] and translation of (acc. number: XP_033706382.1). Amino acid (AA) and gaps (hatched positions) are according to the IMGT unique numbering [65]. AA 91 and 92 are absent in G-ALPHA1 (empty circles (left) or not shown (right)). Positions 61A, 61B, and 72A are characteristic of the G-ALPHA2 and are not reported in the G-ALPHA1 IMGT Collier de Perles. (**B**) IMGT Collier de Perles of 3hg1, available in IMGT/3Dstructure-DB, http://www.imgt.org, accessed on 10 February 2021. The IMGT pMH contact sites [63,64,66,67] between the Homsap G domains and the peptide MLANA (melan-A, MART1) Pr26-35 (ELAGIGILTV A2 > L), also known as melanoma antigen recognized by T cells 1 or MART-1, are displayed with the C1–C11 pMH contact site colors (see also Hs3 in Figure 9).

**Table 1 genes-12-00571-t001:** Characterization of the TRBV subgroups (number of genes, functionality, and CDR-IMGT lengths) in dolphin (*Tursiops truncatus*), goat (*Capra hircus*), pig (*Sus scrofa*), camel (*Camelus dromedarius*), and dog (*Canis lupus familiaris*).

TRBV Subgroup	*Tursiops Truncatus*	CDR-IMGT Lengths	*Capra Hircus*	CDR-IMGT Lengths	*Sus Scrofa*	CDR-IMGT Lengths	*Camelus Dromedarius*	CDR-IMGT Lengths	*Canis Lupus* *Familiaris*	CDR-IMGT Lengths
**TRBV1**	1F	6.6.4	1F	6.6.2	1F	6.6.4	1F	6.6.2	1F	6.6.4
**TRBV2**	1P	-	1F	5.6.4	1F,4P	5.6.4	1F	5.6.4	3P	-
**TRBV3**	-	-	1F	5.6.3	1F	5.6.4	1P	5.6.4	2F,1P	5.6.4
**TRBV4**	1F	5.6.4	1F	5.6.4	4F,1P	5.6.4	-	-	4F	5.6.4
**TRBV5**	3P	5.6.5	18F,12P	5.6.4	2F,1P	5.6.4	3F	5.6.4	2F,2P	5.6.4
**TRBV6**	1P	-	11F,14P4O	5.6.4 5.6.3	1P	5.6.4	1F	5.6.4	1P	-
**TRBV7**	2P	-	2F	5.6.4	2F	5.6.4	2F	5.6.4	1F	5.6.4
**TRBV8**	1O	5.6.4	1P	-	1P	-	1F	5.6.4	1P	-
**TRBV9**	-	-	1P	-	-	-	1P	-	-	-
**TRBV10**	1F	5.6.4	1P	-	1F	5.6.4	1F	5.6.4	1F	5.6.4
**TRBV11**	1P	-	1P	-	1F	5.6.4	1F	5.6.4	1P	-
**TRBV12**	1F	5.6.3	2F	5.6.4	1F,1P	5.6.4	2P	-	1F,1P	5.6.4
**TRBV13**	-	-	-	-	-	-	-	-	-	-
**TRBV14**	-	-	1P	5.6.4	1F	5.6.4	1P	-	-	-
**TRBV15**	-	-	1F	5.6.4	1F	5.6.4	2F	5.6.4	1O	5.6.4
**TRBV16**	-	-	1F	5.6.4	-	-	1F	5.6.4	1F	5.6.4
**TRBV17**	-	-	-	-	-	-	-	-	-	-
**TRBV18**	-	-	1P	-	-	-	-	-	1F	5.6.4
**TRBV19**	1F	-	1F	5.6.4	1F	5.6.4	1F	5.6.4	1P	5.6.4
**TRBV20**	1F	6.7.3	1F	6.7.3	3F	6.7.3	1F	6.7.3	1F	6.7.3
**TRBV21**	-	-	6F	5.6.4 5.5.3	1F	5.6.4	2F,1P	5.6.4	1P	5.6.4
**TRBV22**	1F	5.6.3	1F	5.6.3	1P	-	1F	5.6.4	1F	5.6.4
**TRBV23**	-	-	-	-	1P	-	1P	5.6.4	-	-
**TRBV24**	1F	5.6.4	1F	5.6.4	1F	5.6.5	1P	-	1F	5.6.5
**TRBV25**	-	-	1F	5.6.4	1F	5.6.4	1F	5.6.4	1F	5.6.4
**TRBV26**	1F	5.6.4	1F	5.6.4	-	-	1F	5.6.4	1F	5.6.4
**TRBV27**	1F	5.6.4	1P	5.6.4	1F	5.6.4	1F	5.6.4	1P	5.6.4
**TRBV28**	1P	5.6.4	1F	5.6.4	1F	5.6.4	1F	5.6.4	1F	5.6.4
**TRBV29**	1P	5.6.4	1F	5.6.3	1F	5.7.3	1F	5.7.2	1F	5.7.3
**TRBV30**	1F	6.5.3	1F	6.5.3	1F	6.5.3	1F	6.5.3	1F	6.5.3
**TRBVA**	1P	-	-	-	-	-	-	-	-	-
**Total per Fct**	11F + 11P + 1O		54F + 33P + 4O		27F + 11P		25F + 8P		22F + 1O + 13P	
**Total genes**	23		91		38		33		36	

**Table 2 genes-12-00571-t002:** Interaction energies estimated at the protein interface between TR alpha and TR beta chains and between the reported TR chains and MH1 I-ALPHA.

Pdb	3hg1.pdb		L5RBL13_5RAL11		L5RBL13_5RAL27		L5RBL27_5RAL11		L5RBL27_5RAL27	
**Group1**	A	AB	A	AB	A	AB	A	AB	A	AB
**Group2**	B	C	B	C	B	C	B	C	B	C
**IntraclashesGroup1**	8.81	21.96	10.74	19.12	10.24	18.07	10.51	19.99	8.14	19.16
**IntraclashesGroup2**	12.39	10.93	7.48	13.04	7.24	13.43	8.66	13.22	10.63	15.06
**Interaction Energy**	**−54.67**	**−5.58**	**−21.02**	**−7.23**	**−21.91**	**−5.73**	**−17.32**	**−8.35**	**−17.75**	**−9.26**
**Backbone Hbond**	−10.05	−0.44	−3.95	−0.36	−3.41	−1.94	−3.96	−0.91	−2.91	−2.59
**Sidechain Hbond**	−15.85	−2.94	−9.53	−5.44	−6.10	−4.44	−9.00	−7.16	−6.20	−11.53
**Van der Waals**	−40.11	−11.63	−22.14	−10.44	−19.44	−11.84	−21.66	−11.09	−19.93	−14.02
**Electrostatics**	−6.23	−1.62	−2.43	−4.48	−1.59	−2.17	−3.25	−4.42	−2.07	−2.14
**Solvation Polar**	48.22	16.23	26.43	15.44	22.71	18.22	26.39	17.93	24.38	21.15
**Solvation Hydrophobic**	−53.35	−14.63	−30.35	−11.58	−27.13	−13.32	−29.88	−11.64	−27.49	−14.33
**Van der Waals clashes**	0.63	0.93	0.10	0.80	0.48	1.24	0.24	0.86	0.37	0.56
**entropy sidechain**	15.44	7.15	13.18	7.45	7.76	6.92	15.16	8.16	10.23	11.72
**entropy mainchain**	10.62	1.39	7.84	1.95	5.31	1.97	8.44	0.62	6.74	2.44
**torsional clash**	0.14	0.05	0.80	0.06	0.10	0.13	0.58	0.06	0.01	0.13
**backbone clash**	2.49	2.71	1.35	1.33	0.65	3.45	0.98	1.57	0.72	2.86
**helix dipole**	0.01	−0.01	0.00	0.05	0.00	−0.04	0.00	0.02	0.00	−0.02
**electrostatic kon**	−1.19	−0.05	−0.97	−0.68	−0.60	−0.47	−1.51	−0.78	−0.87	−0.63
**Entropy Complex**	2.38	2.38	2.38	2.38	2.38	2.38	2.38	2.38	2.38	2.38
**Number of Residues**	814	814	657	657	655	655	695	695	693	693
**Interface Residues**	108	44	57	42	52	43	64	40	56	46

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
