# Peer review of "The T Cell Receptor (TRB) Locus in Tursiops truncatus: From Sequence to Structure of the Alpha/Beta Heterodimer in the Human/Dolphin Comparison"

_genes, 2021, doi:10.3390/genes12040571_

Round 1

Reviewer 1 Report

This manuscript completes the analysis of TCR genetics for a dolphin species, the authors having previously published the TCR alpha/delta and beta loci analyses.

This manuscript is of interest to investigators who study TCR genomic evolution and cetacean diseases and health.

The weakness of this paper is that it is written for an audience well versed in Ig and TCR nomenclature, not a very general audience.  The paper could be improved by being less jargony.

Suggestions:

  1. Avoid nomenclature jargon in the Abstract. Save it for being defined in the text. For example I’ve worked on Ig and TCR genes for 30+ years and have no idea what a CDR-IMGT is.  The CDRs as defined using the IMGT system, right?  This is not even well defined in the text itself.  TR-pMH is also a shorthand that should be avoided until defined.

  1. Does the common word “forward” really need to be abbreviated FWD?

  1. Why are some terms in upper case such as STOP-CODON or J-NONAMER? Is this an IMGT nomenclature issue? 

  1. It was a bit confusing that the text describes a conserved artiodactyl three D-J-C clusters and cites Figure 1A but I could only see two D-J-C clusters.

  1. Also referring to the constant region genes and C1, C2 etc and also referring to C1-C11 in the carbon backbone of the domains in the legend of figure 10, was also a bit confusing and maybe should be better explained for clarity.

Reviewer 2 Report

The T cell receptor (TRB) locus in Tursiops truncatus: from sequence to structure of the alpha/beta heterodimer in the human/dolphin comparison
Giovanna Linguiti et al

Reviewer's assessment

The authors extended their prior study (BMC Genomics (2016) 17, Article number: 634) about the genomic and expression analyzes of Tursiops truncatus T cell receptor loci (TRB, TRCA/TRD). However, they now used a basic public sequence repertoire for T cell receptor (TRB) and cDNA analysis from two subjects' blood samples. The typescript structure is molded on the prior study as a template, although the TRB locus analysis is novel. Because the authors used their prior study as a template, the first impression is that the subject is not new, when in fact is. Overall, the analysis is well-executed, and the datasets are robust, making the typescript important for the immunogenetics of the Tursiops truncatus. There are issues with the printed quality of all the figures (as presented), and important points were raised that cause careful amendments.

Material and methods section

  1. Missing prior references for the following tools or databases.
    1. BioProject PRJNA625792: Xiong, Y., Brandley, M.C., Xu, S., Zhou, K., and Yang, G. (2009). Seven new dolphin mitochondrial genomes and a time-calibrated phylogeny of whales. BMC Evol Biol 9, 20. doi: 10.1186/1471-2148-9-20.
    2. The BLAST algorithm for the determination of sequence identity with the corresponding genomic TRBV, TRBD, TRBJ, and TRBC genes (For example NCBI Resource Coordinators (2017). Database Resources of the National Center for Biotechnology Information. Nucleic Acids Res 45, D12-D17. doi: 10.1093/nar/gkw1071).
    3. RepeatMasker tool.
    4. International ImMunoGeneTics information system (see: http://www.imgt.org/about/CitingIMGT.php)
    5. PDB and DeepView - Swiss-PdbViewer.

  1. A chart flow of the overall analysis will facilitate following the Methods section.
  2. Figure 1A, the linear gene map for the 277,931 bp of 1:93263345-93541276 Tursiops truncatus isolate mTurTru1 chromosome 9, mTurTru1.mat.Y should not be broken into two sections. In Figure 1A, the gene boxes should be named according to the nomenclature stated in the typescript.
  3. Figure 1B does not depict all 23 TRBV germline genes described in lines 335-342.
  4. It would be nice if the authors created a public track at the UCSC Genome Browser to map the gene at scale, with the exon-intron structure unveiled. The red-colored boxes on the right side do not match the red/orange colors in the line drawing.
  5. Please note that in line 96, the authors stated that the Tursiops truncatus TRB locus is in forward (FWD, upper strand), but the sense of transcription in figure 1A is antisense (lower strand). This is not very clear.
  6. Figure 1 B; the upper part of panel B, above the sequence alignment, to what exactly does it refer.
  7. Figure 1A awkwardly presents multiple dieresis marks (").
  8. Lines 273-281; 294-298: Figure 1A does not support or depict the comparative structure presented in the text or the D-J-C clusters region's arrangement. A high-resolution drawing is
  9. Lines 284-292: Figure 1A does not support the description in the text. A high-resolution map of the TRY genes is required in a separate figure.
  10. Lines 306-309, about the 12 "RS" preceding sequenced at 5'. A vague statement since all the sequences shown in Figure 2A are preceded by a stretch of over 28 nucleotides (J-NONAMER J-SPACER J-HEPTAMER).
  11. Lines 314-315: The exon-intron structure of the TRBC genes is not apparent (the same applies to all the other gene sets). The precise exon-intron organization must be included in the supplementary tables.
  12. Figures 3, 4, and 7 are printed with poor resolution, insufficient for review.
  13. It is missing the NCBI entries for the 95 sequences used for Figures 3. Please include a supplementary table.
  14. Lines 397-400: Figure 1 does not depict what is described in the text (at the higher resolution required.
  15. Figures 9 and 10 belong to the Results section, not to the Discussion section, and therefore must be transposed. The same applies to the text detailing their significance.

Minot issues:

  1. Spell out AUAP - Abridged Universal Amplification Primer(AUAP)
  2. Lines 343-347 belong to the Methods section.
  3. Line 677: Spleen samples are mentioned, but only peripheral blood was sampled from the two subjects.

Round 2

Reviewer 2 Report

Please see attached assessment file 
